# Harnessing the Power of Reinforcement Learning for Language-Model-Based Information Retriever via Query-Document Co-Augmentation

## Abstract

Recent advances have explored the use of large language models (LLMs) as retrievers by rewriting user queries, while other work has focused on expanding or augmenting corpus documents. However, such unidirectional augmentation, applied to queries or documents in isolation, often fails to reconcile the lexical and stylistic mismatches between them, limiting recall and overall retrieval robustness. To this end, we present an LLM-based retriever empowered to augment both user queries and corpus documents, with its policy fully explored via reinforcement learning (RL) and minimal human inductive bias. Notably, we find that simply training the LLM to augment queries and documents separately, even when combining both at inference, yields little benefit unless paired with our carefully designed bidirectional RL framework, which enables the LLM to simultaneously learn and collaborate on both query and document augmentation policies. A key technical challenge in realizing such a framework lies in jointly updating both policies during training, where the rewards for the two directions depend on each other, making their entangled reward intractable. Our approach addresses this by introducing a reward sampling strategy and a specifically designed RL algorithm that enables effective training with these sampled rewards. Experimental results demonstrate that our approach significantly enhances LLM-based retrieval performance in both sparse and dense settings, particularly in difficult retrieval domains, and achieves strong cross-benchmark generalization. Our code will be publicly released upon acceptance.

## 1 Introduction

Information retrieval (IR) (Baeza-Yates et al., 1999; Singhal et al., 2001) studies the task of finding the best match from a large set of documents based on a given query, and has played a crucial role in recent AI task scenarios, such as retrieval-augmented generation (RAG) (Gao et al., 2023; Wang et al., 2024). Classical IR approaches include sparse retrievals based on TF-IDF (Salton & Buckley, 1988) and BM25 (Robertson et al., 2009), as well as dense retrievals based on embeddings obtained from pre-trained language models (Xiao et al., 2024). With well-established retrievers and the emergence of large language models (LLMs), recent works have identified a bottleneck in poor query quality, thus enhancing IR accuracy through query rewriting (Ma et al., 2023a; Ye et al., 2023; Wang et al., 2023; Shen et al., 2023; Mao et al., 2024). However, retrieval performance still has room for improvement, especially in challenging knowledge domains where accurately retrieving information from a compact corpus is crucial (Dai et al., 2024a;b).

In this work, we argue that unidirectional augmentation—whether applied only to queries or only to documents—fails to fully bridge the lexical and stylistic mismatches between them. Simply enhancing one side in isolation cannot robustly improve retrieval in challenging domains. To address this limitation, we propose an LLM-based retriever that simultaneously augments both queries and documents through collaborative training, thereby pulling challenging queries and documents to be more semantically related and better paired for retrieval. Our augmentation policy is explored purely through reinforcement learning (RL) with minimal human-designed inductive bias.

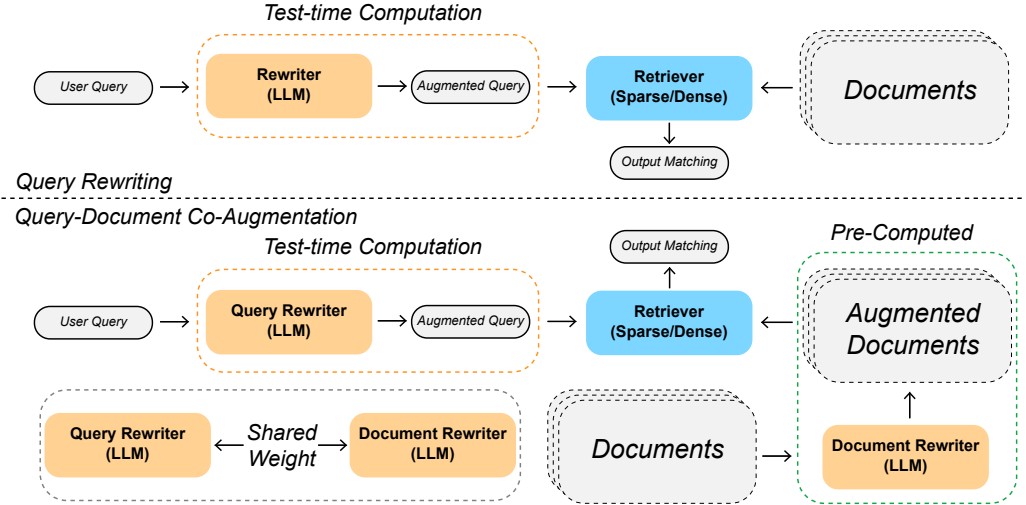

Figure 1: Alongside ordinary test-time query rewriting, we allow the LLM to pre-compute lightweight augmentations for every document, giving it control over both queries and documents. This query–document co-augmentation, when jointly optimized through our bidirectional RL framework, enables the discovery of more accurate retrieval policies especially in challenging collections.

Notably, we find that training the LLM to augment queries and documents separately—even when combining both at inference—yields little benefit unless paired with our carefully designed RL framework to perform joint bidirectional training in a single process, which enables the model to co-operate with itself in both query and document enhancement. A key technical challenge in realizing this training process is the significantly enlarged action space, as the final action is the combination of augmentation actions in both directions, making exact reward computation intractable. We propose a reward sampling strategy for this bidirectional training, and design specialized adjustments to enable the RL algorithm to work with our sampled reward, where direct application of state-of-the-art LLM reinforcement learning algorithms fails to handle our task. To integrate with existing LLM RL frameworks, we adopt a batch-unbatch alternating implementation to achieve effective training under our bidirectional setting.

We conducted experiments on challenging IR benchmarks to verify the efficacy of our approach using both sparse and dense retrievers. The results show that our approach successfully tackled the collaborative training challenge and enabled the model to learn an effective bidirectional augmentation policy, which significantly enhances the performance of both the base model and query enhancement methods. We also provide an analysis of the behavior of the trained policy using our approach, which helps reveal the underlying causes of the improved performance. Furthermore, we observe that the learned LLM-retriever policy achieves desirable cross-benchmark generalization ability, demonstrating that our approach successfully harnesses the power of RL to enable the LLM-retriever to obtain generalizable capabilities in IR through self-exploration.

We provide a detailed discussion of related work in the appendix (see "Related Work").

## 2 PROPOSED APPROACH

### 2.1 OVERVIEW

We propose enhancing a large language model (LLM) via reinforcement learning (RL) to jointly optimize query and document augmentation, rather than treating them as isolated tasks, thereby aligning their word distributions and semantic spaces with the model's internal knowledge for improved retrieval performance. After training, the learned policy can be used to precompute augmented document representations, effectively encoding document knowledge in advance. At inference time,

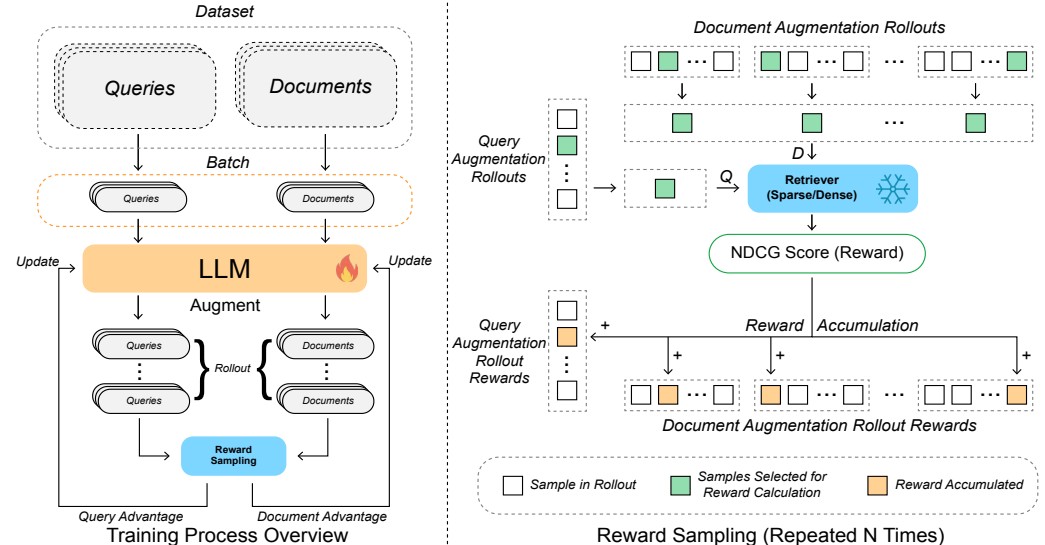

Figure 2: Illustration of the bidirectional RL training process. Left: Overview of the training pipeline. Right: Reward calculation—one sample is randomly selected from the rollouts for each query and related document, its reward is computed, and the reward is accumulated onto the corresponding rollout sample for estimation.

given a user query, the policy augments both the query and performs retrieval over the preprocessed document collection with computational costs comparable to standard query rewriting. Since query and document augmentation are formulated as text processing tasks, they remain independent of the underlying retrieval method, enabling flexible integration with various retrieval modules (e.g., BM25 (Robertson et al., 2009) for sparse retrieval or BGE models (Xiao et al., 2024) for dense retrieval).

The overview of the training pipeline is illustrated in Fig. 2. To enable joint training of query and document augmentation, we introduce a novel query-document composite sampling strategy (Sec. "Query-Document Composite Sampling"). This approach organizes queries and their associated documents into the same batch during sampling, effectively reducing the training data scale from the entire document collection to a manageable batch size. For reward computation (Sec. "Within-Batch Reward Computation"), we perform retrieval by executing query rollouts within document rollouts for each batch, using the average score of each rollout as the reward signal. Owing to the distinct characteristics of our reward computation, conventional group-wise or batch-wise reward normalization methods, such as those used in GRPO (Guo et al., 2025) or REINFORCE++ (Hu, 2025b), are not directly applicable. Therefore, we adapt the advantage computation method (Sec. "Taming Reward Variance in RL") to better align with our task requirements. Notably, despite introducing more sophisticated sampling, reward computation, and advantage calculation procedures, our approach remains fully compatible with standard LLM RL training pipelines (Sec. "Batch-Unbatch Alternating Implementation"). Specifically, rollout inference and backpropagation are performed at the individual text level, while sampling, reward computation, and advantage calculation are conducted at the group level.

## 2.2 QUERY-DOCUMENT COMPOSITE SAMPLING

To enable synchronous augmentation training for both queries and documents, a straightforward approach would be to evaluate lexical and semantic alignment by performing retrieval with augmented queries over the augmented document collection. However, executing inference at the scale of the entire document collection during every training step is computationally infeasible. To address this, we modify the sampling strategy to preserve the original batch size while ensuring that each batch contains three essential components: queries, relevant documents, and irrelevant documents, thereby forming a representative mini-dataset.

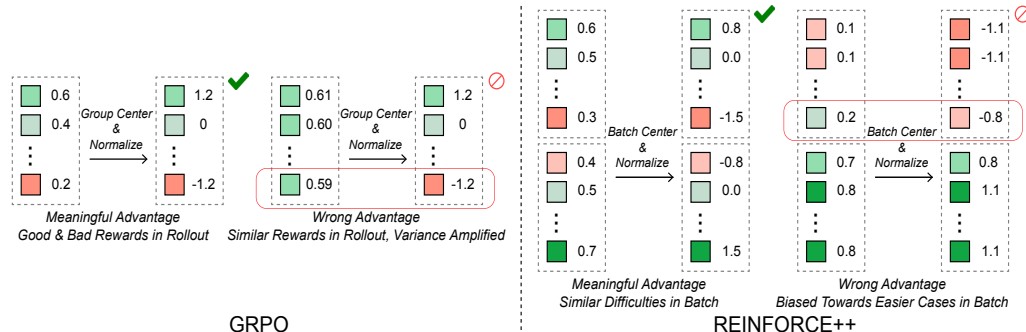

Figure 3: Illustration of the problems identified in our mission when adopting state-of-the-art reinforcement learning algorithms. The unique challenge of our task requires specialized design.

The sampling procedure is as follows: We first randomly select $q$ queries from the query pool. For each query, we sample $d_{pos}$ documents with positive retrieval scores (i.e., relevant documents). Additionally, we sample $d_{neg}$ documents (irrelevant documents) that have zero retrieval scores with respect to all $q$ queries. This combination of queries, relevant documents, and irrelevant documents constitutes a complete data group. Each batch thus contains $q + q \times d_{pos} + d_{neg}$ distinct texts, which are assigned appropriate system prompts based on their type (query or document) before being processed by the text augmentation model as training data.

## 2.3 WITHIN-BATCH REWARD COMPUTATION

Given the inherent semantic and lexical disparities between user queries and documents, employing a single critic model to evaluate augmented texts originating from different distributions may result in suboptimal assessment quality. Following the methodology of GRPO, we generate $n_{rollout}$ augmented rollouts for each text and compute rewards for each variant at the group level. However, the combinatorial complexity increases exponentially with the number of documents: for a batch containing $q + q \times d_{pos} + d_{neg}$ texts, evaluating all possible configurations would require processing: $q \times n_{rollout} \times n_{rollout}^{d_{pos}+d_{neg}}$ matching pairs, which is exponential in the number of documents.

We observe that while query rollouts remain independent, each combination of a document's $n_{rollout}$ augmented rollouts introduces potential variations in similarity rankings during retrieval, which consequently influences reward computation. To address this challenge, we adopt a straightforward multi-sampling strategy applied to the mini-batch of documents.

Specifically, in each sampling iteration, we randomly select one rollout for each document and then evaluate all query rollouts by calculating their NDCG scores (Järvelin & Kekäläinen, 2002) based on the similarity rankings obtained from retrieval. These scores serve as rewards for the selected combinations of query and document rollouts. The final reward for each rollout is calculated by averaging between all sampling iterations. This sampling method incurs significantly lower computational overhead (less than $1e^{-1} \times$ inference time) while accurately estimating the rewards for each query and document rollout (error $\leq 1e^{-2}$), thereby enabling efficient and effective synchronous training of both query and document augmentation components.

## 2.4 TAMING REWARD VARIANCE IN RL

While the sampling strategy serves as an effective reward estimator, it inevitably introduces some variance. However, our within-batch reward computation can easily reduce this variance to the level of $1e^{-5}$, rendering reward fluctuations negligible for policy learning in most cases. In GRPO, the advantage for each rollout is computed as $(r - r_{mean})/r_{std}$. As illustrated in Fig. 3, we observe that when all rollouts within a group receive identical rewards, the within-group normalization in GRPO amplifies the originally negligible sampling variance to 1. This leads to random advantage assign-

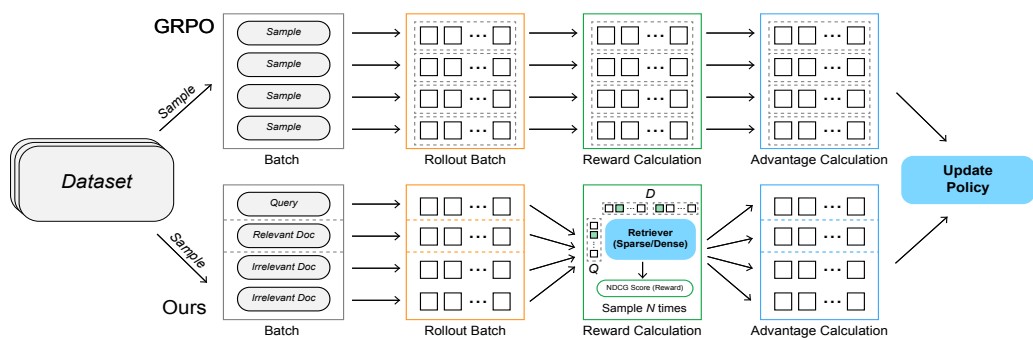

Figure 4: Implementation of our RL training pipeline. The entanglement is introduced by our reward calculation. Thus, we pack queries and documents into a single batch, and perform paralleled sampling for rollout efficiency. We only entangle the contents for reward computation, and revert them to separate samples for advantage computation and policy update.

ments among rollouts, without reflecting meaningful quality differences. Consequently, substantial noise is introduced into the optimization process, ultimately resulting in training failure.

We subsequently investigated REINFORCE++, which modifies the GRPO approach by replacing within-group reward normalization with batch-wide normalization. However, in our experimental setup, significant variations in average scores attributable to differing query difficulties introduce a critical limitation. As shown in Fig. 3, REINFORCE++'s batch-wide normalization causes the advantage computation to be dominated by query difficulty rather than the quality of text augmentation. As a result, rollouts from the same sample often become uniformly positive or negative, which leads to poor training performance.

Returning to the fundamentals of reinforcement learning training, our primary concerns regarding advantage computation are numerical stability and discriminative power, rather than standardization. Notably, our implementation of expected NDCG scores as rewards (as detailed in Sec. "Within-Batch Reward Computation") naturally satisfies these requirements. These scores maintain stable distributions within the $[0, 1]$ range while effectively capturing variations in augmentation quality. Our final solution is simple but effective. We remove within-group normalization while retaining within-group centering. This modification successfully eliminates noise arising from identical rewards, while preserving the original reward differentials among group rollouts, resulting in substantial improvements in training performance. Additionally, we apply differential scaling to the advantage scores computed from queries, relevant documents, and irrelevant documents to balance their contributions during training. This adjustment mitigates the risk of gradient domination by the disproportionately large number of irrelevant documents.

To validate these findings, we conducted comprehensive ablation studies comparing various approaches. The respective impacts on training efficacy are discussed in detail in Sec. "Analysis on Normalization Techniques".

## 2.5 BATCH-UNBATCH ALTERNATING IMPLEMENTATION

To align with conventional RL training frameworks, we adopt a batched-unbatched alternating paradigm shown in Fig. 4. The core distinction from previous approaches lies in our batch construction, where individual samples are intrinsically interlinked rather than independently distributed.

**Batch-Level Sampling**. We redesigned the dataset sampling mechanism to generate batches comprising queries, relevant documents, and irrelevant documents, following the methodology described in Sec. "Query-Document Composite Sampling". Each text sample is augmented with system prompts according to its source before being assembled into the same batch. This design ensures the integrity of each batch, allowing it to be used independently for training.

**Sample-Level Inference**. During model rollout operations, batched text samples are logically unbatched into individual samples. At this stage, the distinct sources of the samples (queries or doc-

uments) become transparent to the inference engine, enabling parallel processing of all rollouts without mutual interference. Importantly, this phase maintains complete compatibility with existing RL training infrastructures.

**Batch-Level Reward Computation**. All rollouts are reaggregated at the batch level for reward calculation. We group the selected rollouts based on their sources (queries or documents), and then apply the retrieval module to rank them and compute the corresponding rewards. Subsequently, group-wise centralization is performed for each sample to provide advantage estimates for individual rollouts.

**Sample-Level Parameter Update**. After calculating the advantages, the text samples are unbatched once again. Although processed in batch form, policy gradient losses are individually computed for each rollout using their respective advantage values, and the model parameters are then updated accordingly. This step is fully compatible with existing reinforcement learning training frameworks and does not require additional modifications.

## 3 EXPERIMENTS

We conduct experiments to evaluate the effectiveness of our proposed reinforcement learning framework, as well as to investigate the importance of query-document collaborative training. The experimental results substantiate the core motivations underlying our training framework and demonstrate its performance advantages. Additionally, we perform ablation studies to examine the impact of different advantage calculation strategies on model performance.

### 3.1 EXPERIMENTAL SETUP

We adopt the BEIR benchmark (Thakur et al., 2021) and select three of its datasets for model training. These datasets are used to evaluate the benefits of in-domain training and to assess model generalization through cross-benchmark testing. Training LLMs using RL is computationally expensive. Due to resource constraints, training is limited to a maximum of 300 steps per dataset, and we choose to allocate our resources to experiments centered around one base model, Qwen2.5-7B (Qwen et al., 2025), for a fair evaluation of improvements.

To investigate the impact of query-document collaborative training, we conduct ablation experiments on the NFCorpus dataset. Specifically, we train models under three settings—query-augmentation-only, document-augmentation-only, and query–document collaborative augmentation—and evaluate their individual and combined contributions to retrieval performance. Furthermore, we compare the effects of different advantage calculation methods via an additional ablation study. We compare existing normalization strategies from GRPO and REINFORCE++, as well as our centralization-only setting. All experiments adopt NDCG@10 (Järvelin & Kekäläinen, 2002) as the evaluation metric.

Notably, our framework is designed to be orthogonal to the underlying retrieval architecture, supporting flexible integration with different retrieval modules. To demonstrate this, we evaluate our method in both sparse and dense retrieval settings. For sparse retrieval, we adopt BM25 as the base retriever, where the LLM is prompted to first summarize textual content and then generate query expansions or document keywords in the form of discrete words. For dense retrieval, we utilize BGE-base-en-v1.5, where the LLM produces sentence-level query expansions or document summaries. These augmented outputs are concatenated with the original contents, aligning with the retrieval mechanisms: BM25 relies on discrete word matching, while BGE-base-en-v1.5 uses holistic paragraph understanding and vectorization.

### 3.2 PERFORMANCE ON SPARSE AND DENSE RETRIEVAL

Tab. 1 presents a comprehensive comparison of NDCG@10 performance across five datasets under both sparse (BM25) and dense (BGE-base-en-v1.5) retrieval settings. The results include various model scales and training strategies, with the best and top-three results highlighted.

Our training method consistently outperforms baseline retrievers across different datasets and retrieval settings. Notably, Ours-7B achieves the best results on all datasets under the sparse setting and

Table 1: NDCG@10 Performance under sparse (BM25) and dense (BGE-base-en-v1.5) retrieval. The best results are bolded, and other top-three results are underlined.

| Settings | | NFCorpus | SciFact | FiQA-2018 | SCIDOCS | TREC-COVID |
|---|---|---|---|---|---|---|
| Base Retriever | BM25 | 0.343 | 0.691 | 0.254 | 0.165 | 0.688 |
| | BGE-base-en-v1.5 | 0.368 | 0.738 | 0.391 | 0.214 | 0.672 |
| Qwen2.5-3B | BM25 | 0.352 | 0.692 | 0.237 | 0.155 | 0.642 |
| | BGE-base-en-v1.5 | 0.363 | 0.741 | 0.359 | 0.210 | 0.772 |
| Qwen2.5-7B | BM25 | 0.363 | 0.696 | 0.258 | 0.162 | 0.694 |
| | BGE-base-en-v1.5 | 0.371 | 0.746 | 0.383 | 0.212 | 0.776 |
| QwQ-32B | BM25 | 0.361 | 0.709 | 0.276 | 0.162 | 0.642 |
| | BGE-base-en-v1.5 | **0.391** | **0.769** | 0.373 | 0.223 | 0.756 |
| Qwen2.5-72B | BM25 | 0.370 | 0.747 | 0.305 | 0.164 | 0.670 |
| | BGE-base-en-v1.5 | 0.377 | 0.751 | **0.395** | 0.220 | 0.734 |
| Ours-3B | BM25 | 0.371 | 0.715 | 0.273 | 0.168 | 0.716 |
| | BGE-base-en-v1.5 | 0.379 | 0.749 | 0.364 | 0.217 | 0.771 |
| Ours-7B | BM25 | **0.403** | **0.748** | **0.328** | **0.181** | **0.727** |
| | BGE-base-en-v1.5 | 0.384 | 0.753 | **0.395** | **0.224** | **0.807** |

Table 2: NDCG@10 performance of query-only and doc-only ablation studies on NFCorpus. Base-Q and Base-D represent using the base model (Qwen2.5-7B) to enhance only queries or only documents, respectively. RL-Q and RL-D refer to models trained using RL for query-only augmentation and document-only augmentation, respectively. The plus sign (+) indicates that the methods are used jointly. RL-QD refers to collaborative training for bidirectional augmentation (our proposed method).

| Settings | Base Retriever | Base-Q | Base-D | Base-Q + Base-D | RL-Q | RL-D | RL-Q + RL-D | RL-QD (ours) |
|---|---|---|---|---|---|---|---|---|
| BM25 | 0.343 | 0.357 | 0.356 | 0.363 | 0.381 | 0.372 | 0.388 | **0.403** |
| BGE-base-en-v1.5 | 0.368 | 0.377 | 0.364 | 0.371 | 0.379 | 0.373 | 0.372 | **0.384** |

most datasets under the dense setting, surpassing even the much larger models like Qwen2.5-72B and QwQ-32B. Ours-3B also demonstrates significant improvements over its base model, though its performance still lags behind larger models. These results indicate that our approach provides substantial enhancements to base models of varying scales.

Examining the cross-domain generalization capability, our models, trained on individual datasets, maintain strong performance when evaluated on unseen domains such as SCIDOCS and TREC-COVID. In particular, under the dense retrieval setting on TREC-COVID, our approach achieves a 13% improvement over the base retriever and further outperforms the base model with augmentation by 3.1%.

We also observe that, for training-free models, retrieval performance generally improves with model size, although this trend is not strictly monotonic. In some cases, larger models do not yield better augmentation or retrieval results, likely due to misalignment between independently generated query and document augmentations. Furthermore, for smaller models such as Qwen2.5-3B, the limited instruction-following capability may result in suboptimal augmentation outputs, such as empty strings or outputs in incorrect formats, thereby constraining the benefits of our training method.

**Cross-benchmark generalization**. We conducted cross-benchmark validation on the models trained on three different datasets to evaluate the generalization capability of our method. As observed, in the sparse retrieval setting, our models demonstrated strong generalization ability: models trained on any single dataset achieved notable performance improvements over the untrained Qwen2.5-7B when applied to unseen domains. In contrast, under the dense retrieval setting, the generalization performance varied across domains. While improvements were observed on NFCorpus and SciFact, the performance on FiQA-2018 was inferior to that of Qwen2.5-7B.

Table 3: Cross Entropy of query-only and doc-only Ablation Studies on NFCorpus.

| Settings | | H(Q, D) |
|---|---|---|
| Qwen2.5-7B | BM25 | 10.318 |
| | BGE-base-en-v1.5 | 8.314 |
| queryonly | BM25 | 9.998 |
| | BGE-base-en-v1.5 | 8.269 |
| doconly | BM25 | 10.096 |
| | BGE-base-en-v1.5 | 8.467 |
| Ours | BM25 | **9.501** |
| | BGE-base-en-v1.5 | **7.743** |

Table 4: NDCG@10 Performance of Config Ablation Studies on NFCorpus.

| Settings | | NDCG@10 |
|---|---|---|
| Qwen2.5-7B | BM25 | 0.363 |
| | BGE-base-en-v1.5 | 0.371 |
| groupnorm (GRPO) | BM25 | 0.376 |
| | BGE-base-en-v1.5 | 0.374 |
| batchnorm (RF++) | BM25 | 0.364 |
| | BGE-base-en-v1.5 | 0.354 |
| w/o adv scale | BM25 | 0.366 |
| | BGE-base-en-v1.5 | 0.348 |
| Ours | BM25 | **0.403** |
| | BGE-base-en-v1.5 | **0.384** |

## 3.3 Essentialness of Query-Document Co-Augmentation

Tab. 2 shows ablation studies on query-document co-augmentation. Models trained with only query or document augmentation, whether used alone or even combined at inference, exhibit a substantial performance gap compared with our collaboratively trained model. This indicates that single-direction augmentation—applied to queries or documents in isolation—fails to effectively reconcile lexical and stylistic mismatches, limiting retrieval performance.

The superior performance of our approach arises from the collaborative bidirectional training, which enables the LLM to simultaneously learn and align both query and document augmentation policies. Only through this joint training can the model effectively match queries and documents semantically, resulting in significant improvements across sparse and dense retrieval settings.

To further investigate the detailed behavior of our explored collaborative bidirectional augmentation policy, we conduct qualitative and quantitative analyses of the augmented results. We first compute the word distributions of the augmented queries and documents across the entire dataset for all models used in the ablation study of Tab. 2, and calculate the cross-entropy $H(Q, D)$ of the query distribution relative to the document distribution, which quantitatively reflects the behavior of the output token space given the same input. As shown in Tab. 3, the cross-entropy for models trained with collaborative query-document augmentation is substantially lower than that of other models in both sparse and dense retrieval settings. This supports our motivation in enhancing both queries and documents: collaborative training enables the policy to learn to cooperate with itself to match queries and documents in the semantic domain, thereby improving retrieval effectiveness, which is only possible when trained in a bidirectional manner. With the policy understanding how to align the queries and documents instead of overfitting to the knowledge of a specific corpus, this explains the observed generalization across datasets, which can be essential when deployed to a new corpus. We also note that the generalization ability from the BM25 model is stronger than that of the policy explored using the dense retrieval model. This might be caused by dense retrievers developing implicit representations with domain preference (Zhao et al., 2024b), which is also observed and reported in our concurrent work on RL-based query augmentation (Jiang et al., 2025).

**Case study**. We also provide qualitative analysis with a case study, focusing on analyzing sparse retrievers, as the retriever operates in word space, which is directly interpretable by humans, as presented in Fig. 5. In the top case, we aimed to retrieve a target document concerning "(210)Po" using the query "carcinogens". It can be observed that all models were able to generate relevant words during the augmentation process that semantically "bridge" the gap between the query and the document. For example, words such as "DNA", "carcinogenesis", "genetic mutation", and "chemical exposure" were generated to augment the query, while words like "radiation" and "radioactivity" were generated to augment the document. However, these semantically related words failed to achieve a successful match due to discrepancies in word distribution. In contrast, our proposed method augments both the query and the document with a consistent word distribution, resulting in the presence of shared words such as "radiation" and "risk" in both augmented texts. This lexical alignment facilitates the successful retrieval of the target document. In the bottom case, our method expands the query "Probiotics" with related words such as "Gut Health", "Microbiome", "Supplements", "Di-

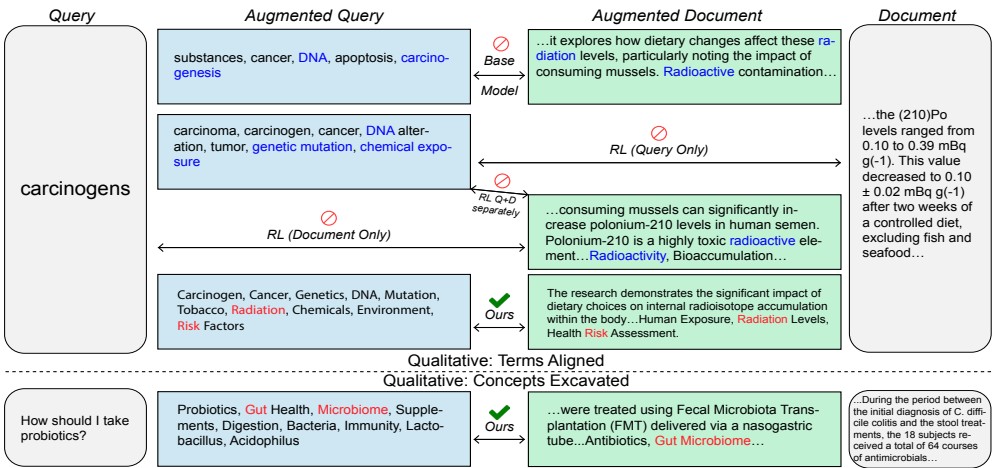

Figure 5: Qualitative demonstration of RL behaviors that boost performance. *Top:* Concepts are discovered, but terms are not unified. Related terms are marked in blue, while matched terms are marked in red. The base model, query-only, and document-only models all discovered related concepts, but with misaligned terms. Enhancing the documents alone does not directly improve performance; instead, success arises from joint bidirectional training. *Below:* We show another case where our policy also learns to discover concepts to retrieve related documents, whereas other methods failed to generate related concepts (failed results omitted for conciseness).

gestion", "Bacteria", "Immunity", "Lactobacillus", and "Acidophilus", and augments the document mentioning "treated using Fecal Microbiota Transplantation (FMT)" with words related to the "Gut Microbiome", thus successfully retrieving a target document with otherwise low textual overlap.

### 3.4 ANALYSIS ON NORMALIZATION TECHNIQUES

Finally, we conduct ablation studies on the advantage calculation settings described in Sec. "Taming Reward Variance in RL" to highlight the performance benefits of our approach. Tab. 4 compares the results of group normalization (GRPO), batch normalization (REINFORCE++), and centralization only (ours). Group normalization amplifies sampling variance, introducing excessive noise and impairing optimization. Batch normalization disrupts intra-group reward comparisons, causing advantage calculations within a batch to be dominated by query difficulty, resulting in negligible training effectiveness; in dense retrieval, it even underperforms direct augmentation with Qwen2.5-7B. In contrast, our centralization-only setting preserves intra-group reward relationships and thus achieves the best performance. Table 4 also demonstrates the necessity of advantage scaling after centralization: without it, the training direction is dominated by the majority of irrelevant documents, which constitute the majority within each batch, leading to poor performance.

## 4 LIMITATIONS AND FUTURE WORK

While our collaborative bidirectional RL framework significantly improves retrieval performance, training becomes challenging when the number of candidate documents is very large. Although this is less of a concern during testing—since augmented documents can be precomputed—the primary computational bottleneck arises during training, where both query and document augmentation policies must be updated jointly. Our method introduces sampling techniques to reduce the cost of generating the substantially increased rollouts required in this setting.

These considerations suggest that our approach is particularly well-suited to semantically complex domains with moderately sized corpora, where bidirectional training can enhance retrieval effectiveness without incurring prohibitive costs. We believe that this leads to important and meaningful future work in applying our approach to challenging IR applications. Moreover, our RL framework may offer insights for reinforcement learning scenarios that rely on jointly computing rewards across interdependent samples.

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

# A    RELATED WORK

With the wide adoption of LLM systems, information retrieval (IR) systems are gaining increasing importance through retrieval-augmented generation (RAG), reducing hallucination and enriching factual knowledge (Mallen et al., 2022; Shi et al., 2023; Chen et al., 2017; Lee et al., 2019; Guu et al., 2020; Lewis et al., 2020; Lazaridou et al., 2022; Asai et al., 2024). IR systems also benefit from LLMs to increase their retrieval accuracy by rewriting queries (Ma et al., 2023a), as the queries might not provide sufficient information that can be faithfully related to the document, hampering IR accuracy (Dageville et al., 2016; Belkin et al., 1982). In this work, we focus on enhancing the ability of LLMs to improve IR accuracy using reinforcement learning. We will first discuss retrieval augmentation methods, then discuss methods for enhancing LLMs in IR tasks.

## A.1    RETRIEVAL AUGMENTATION

Previous methods mainly enhance information retrieval tasks by developing better retrievers (Chen et al., 2017; Karpukhin et al., 2020), enhancing retrievers and readers together (Karpukhin et al., 2020; Lewis et al., 2020; Sachan et al., 2021; Lee et al., 2022; Jiang et al., 2022), or by enhancing queries with external knowledge (Zesch et al., 2007; 2008; Syed, 2010; Dalton et al., 2014; Xu et al., 2009; Meij et al., 2010; Xiong & Callan, 2015) and relevant content (Abdul-Jaleel et al., 2004; Metzler & Croft, 2005; 2007).

In the era of LLMs, as sparse retrievers are well-established and tuning dense retrievers requires a lot of data to reduce overfitting (Ma et al., 2023a), recent methods mainly focus on leveraging or improving the ability of LLMs in information retrieval (Trivedi et al., 2023; Yao et al., 2023; Khattab et al., 2022; Press et al., 2022). Studies have revealed that, pre-trained on large corpora, LLMs without fine-tuning already serve as powerful query optimizers (Shen et al., 2023; Wang et al., 2023; Brown et al., 2020; Touvron et al., 2023).

## A.2    REINFORCEMENT LEARNING FOR ENHANCING LLMS IN IR

With the success of RLHF in aligning LLMs with human preferences (Christiano et al., 2017; Stiennon et al., 2020; Ouyang et al., 2022), reinforcement learning has emerged as a principled approach for enhancing LLMs, such as PPO (Schulman et al., 2017), as well as recent methods including GRPO (Guo et al., 2025) and REINFORCE++ (Hu, 2025a), which have demonstrated significant performance gains in tasks by exploring based on reward beyond preference alignment (Guo et al., 2025).

In the domain of information retrieval, recent methods have also explored utilizing reinforcement learning to improve query augmentation. Some systems leverage reward feedback from the final search or generation results (Ma et al., 2023b; Fan et al., 2024; Zhao et al., 2024a). Another line of work, which is most relevant to us, directly utilizes feedback signals, including a recent work exploring trial-and-error from metric feedback (Hsu et al., 2024) and a concurrent work (Jiang et al., 2025) utilizing feedback signals to enhance IR tasks and SQL tasks. Our work also focuses on improving the performance of LLMs in the specific IR task. The main difference is that our proposed method enables the LLM to explore a policy that not only augments the query but also manages the document itself simultaneously, which significantly improves performance in challenging IR domains.

# B    ADDITIONAL EXPERIMENTAL RESULTS

## B.1    RESULTS OF STATISTICAL EVALUATION ON NORMALIZATION TECHNIQUES

In this section, in addition to the empirical evidence from the ablation study, we also present statistical evidence to support the problem analysis in Sec. "Taming Reward Variance in RL" and Sec. "Analysis on Normalization Techniques". The experiments are conducted using the same settings as our ablation studies.

Specifically, we collect statistical evidence for the two problems illustrated in Fig. 3: *Variance Amplified* and *Biased Towards Easier Cases*. For a clear quantitative demonstration, we define two

| Settings | | Amplified Variance | Same Sign |
|---|---|---|---|
| groupnorm (GRPO) | BM25 | 94.4% | 0.0%* |
| | BGE-base-en-v1.5 | 94.8% | 0.0%* |
| batchnorm (RF++) | BM25 | 19.1% | 63.9% |
| | BGE-base-en-v1.5 | 7.5% | 84.4% |
| Ours | BM25 | 0.0%* | 0.0%* |
| | BGE-base-en-v1.5 | 0.0%* | 0.0%* |

Table 5: Anomalous group proportion under different normalization settings. "Amplified Variance" means the variance in reward is significantly amplified after normalization. "Same Sign" means all advantages in the group are positive or negative. "*" indicates this problem will not occur in theory. For more details, please see Sec. "Results of Statistical Evaluation on Normalization Techniques"

anomaly indicators and report their occurrences. To detect *amplified variance*, we track the original percentage of groups with variance below a threshold $std_{threshold}$, and subtract the corresponding percentage after normalization. To detect *biased advantage*, we evaluate the occurrence of all advantages in a group sharing the same sign (all positive or all negative), which strongly indicates bias introduced by batch normalization.

As shown in Tab. 5, we observe that in our task, GRPO and our method do not exhibit the biased advantage reflected by the occurrence of the "same sign" phenomenon, as there is no batch-based advantage calculation. Regarding amplified variance, we choose a small threshold, and the resulting statistics demonstrate two key points: 1) there is initially a large proportion of rewards with small variance below our set threshold, and 2) these are amplified after normalization, especially in GRPO, as values below 0.02 are unavoidably amplified to 1. The selection of the threshold does not affect our conclusion, as it clearly shows that many rewards have very small variances, which are greatly amplified. We believe the abundance of small-variance groups is due to the fact that augmentation does not necessarily alter the similarity ranking in retrieval, especially for documents, where each batch contains many randomly irrelevant documents whose augmented results are unlikely to affect the reward. This also indicates that the variance introduced by our reward sampling computation is not significant, and that the problem is caused by subsequent computation rather than the variance of the reward sampling itself. While batch normalization reduces this problem, it causes many groups to have only positive or negative advantages, which does not represent a meaningful direction of optimization. In contrast, by only performing centering and allowing the variance to be naturally controlled by the reward computation itself, we avoid amplifying sampling errors in the reward and do not introduce intra-group bias, enabling effective reinforcement learning and successfully improving performance.

## C  IMPLEMENTATION DETAILS

### C.1  TRAINING SETUP

We adopt a GRPO-based training pipeline, following most hyperparameters from the countdown experiment in TinyZero (Pan et al., 2025), with several modifications to accommodate the requirements of different tasks. Specifically, the temperature is set to 1.2 to encourage exploration of diverse augmentations. A repetition penalty of 1.2 is applied to prevent the model from becoming trapped in locally optimal solutions that simply copy the original text. The micro batch size is set to 16, with each micro batch containing one query and 1–5 relevant documents (depending on the dataset), while the remaining slots are filled with randomly selected irrelevant documents. The batch size is set to 512 and the mini batch size to 128 to ensure gradient stability. We select advantage scale coefficients of 1.0, 0.2, and 0.1 for queries, relevant documents, and irrelevant documents, respectively, to balance the proportions of each component. Additionally, we remove the KL loss and entropy loss terms, as these constraints hinder further model optimization, which is consistent with findings reported in concurrent work DAPO (Yu et al., 2025). The format reward is also omitted; however, we still extract augmented content from within ¡answer¿¡/answer¿ tags, defaulting to empty aug-

mentation if such tags are absent. Under this configuration, the model quickly learns to follow the required format while avoiding performance degradation due to excessive adherence to formatting.

## C.2 Environment Setup

We utilize VERL (Sheng et al., 2024) as the LLM RL fine-tuning framework, building upon TinyZero. The software packages and runtime environment are configured to be compatible with this version of the training framework, including Python (v3.9), CUDA (v12.4), VLLM (v0.6.3) (Kwon et al., 2023), PyTorch (v2.4.0), and Ray (v2.46.0) (Moritz et al., 2018). For sparse retrieval and BM25 evaluation, we employ ElasticSearch (v7.10.2), while FAISS-GPU (v1.7.2) (Douze et al., 2024) is used to support efficient dense vector matching. All experiments are conducted on a single node equipped with eight NVIDIA A100 80GB GPUs.

# D The Use of Large Language Models

## D.1 Research subject

We develop an LLM-based retriever that jointly augments both queries and documents, and design a bidirectional RL framework to optimize these augmentation policies collaboratively.

## D.2 Writing assistant

We utilize LLMs to help proofread the manuscript and fix writing issues.

