# OpenReview forum: "Harnessing the Power of Reinforcement Learning for Language-Model-Based Information Retriever via Query-Document Co-Augmentation"
_ICLR.cc/2026/Conference — ICLR 2026 Conference Withdrawn Submission_

### Official Review · Reviewer_Xr3N · 2025-10-27

**Soundness:** 2
**Presentation:** 1
**Contribution:** 3
**Rating:** 2
**Confidence:** 5

**Summary:**

The paper proposes a reinforcement learning framework for enhancing large language models (LLMs) in information retrieval (IR). It introduces a query–document co-augmentation method, enabling the LLM to simultaneously augment both the query and relevant documents. This bidirectional optimization is designed to improve retrieval effectiveness across sparse and dense retrieval architectures (e.g., BM25, BGE-base-en-v1.5).

**Strengths:**

The paper introduces a well-motivated reinforcement learning framework that jointly optimizes both query and document representations, rather than treating them as separate components.
The experiments are well-structured, spanning both sparse (BM25) and dense (BGE-base-en-v1.5) retrieval backbones.

**Weaknesses:**

Overall, the paper’s idea is interesting, but the presentation is vague and lacks clarity, especially in mathematical formulation and experimental description. The following issues should be addressed to improve readability and credibility.

**Clarity**
1. The paper does not provide clear mathematical notations when describing the reward function and other key components of the framework. This makes the overall process vague and hard to follow. The authors are encouraged to formally define all important equations (e.g., reward computation) to improve clarity.
2. The prompts used during RL training are not provided. This information is important because prompts determine the exploration direction of LLMs during RL training. The paper should include the exact prompts or at least representative examples.
3. (Line 294) – The authors mention “3 of the datasets” from BEIR, but do not specify which ones are used. Please clearly indicate which datasets are used for training and which for evaluation.
4. (Lines 407–410) – The computation process described here should include explicit mathematical notations. The current textual explanation is vague to understand how the calculations are performed.

**Accuracy for presentation**
1. (Lines 248–249) – The sentence “remove within-group normalization while retaining within-group centering” actually corresponds to Dr. GRPO (see Dr. GRPO, https://arxiv.org/pdf/2503.20783). The authors should cite this work accordingly.
2. (Line 370) – The statement that Qwen2.5-3B’s instruction capabilities are limited is not accurate. According to Qwen2.5 technical report, https://arxiv.org/pdf/2412.15115, its instruction-following ability is relatively strong. The suboptimal augmentation outputs are more likely due to the smaller capacity of Qwen2.5-3B, leading to a narrower exploration space compared with Qwen2.5-7B. As a result, Qwen2.5-3B cannot sample better trajectories during RL training.
3. (Figure 3) – The caption’s phrase “state-of-the-art reinforcement learning algorithms” is not accurate. It would be more accurate to say “representative reinforcement learning algorithms for optimizing LLMs,” and you should emphasize that these algorithms are representative within the LLM + RL setting, to narrow the scope of whole RL research.
4. (Line 788) – The phrase “based on reward beyond preference alignment” is inaccurate. In RL training, preference alignment is also a form of reward signal. Thus, the distinction made here is conceptually incorrect and should be rephrased for precision.

**Experimental Limitations**
1. The paper does not include comparisons with recent strong baselines such as DeepRetrieval [https://arxiv.org/pdf/2503.00223]. Although some baselines are mentioned in the related work section, experimental comparisons are necessary to demonstrate relative performance.
2. In Figure 5, it is unclear why the “original document” does not contain the term “Radioactive.”, since my understanding is this doc should be relevant to "Radioactive", and the doc should have mentioned the keywords in the document. This should be clarified to help readers interpret the qualitative results properly.

**Minors**
1. Line 863  ¡answer¿¡/answer¿ -> < answer > < /answer >, some rendering problems
2. Line 462, calculation settings described in Sec -> Sec [what?]

**Questions:**

See the weaknesses.

---

> ### Author Response · Authors · 2025-11-26
>
> We sincerely thank the reviewer for their detailed feedback and for acknowledging the contribution of the bidirectional RL framework for query-doc collaborative augmentation across both sparse and dense retrieval architectures.
>
> Below, we provide detailed responses to the points raised. R? means the response to Q? in "Weaknesses", and A? means the response to Q? in "Questions".
>
> **R1.**
> To clarify, we use within-group recall measured by NDCG as the rule-based reward. For example, for NDCG@10, the reward is computed as follows:
>
> $$\text{NDCG@10} = \frac{1}{Z} \sum_{i=1}^{10} \frac{2^{rel_i} - 1}{\log_2(i+1)}$$
>
> where $rel_i$ denotes the relevance of the $i$-th retrieved document, and $Z$ is the normalization constant for the ideal ranking. Detailed implementation can be found in our codebase.
>
> **R2.**
> We thank the reviewer for pointing this out.
> As mentioned in our R2 to Reviewer VBdL, we will include all prompts used for both query and document augmentation in the supplementary material.
>
> **R3.**
> Regarding the clarification in Line 294, the three BEIR datasets used for RL training are NFCorpus, SciFact, and FiQA-2018. The other datasets (SCIDOCS and TREC-COVID) mentioned in our experiments are strictly used for evaluation only, with no tuning or training performed on them.
>
> **R4.**
> Regarding the clarification in Line 407-410， when computing the effect of augmenting queries and documents with different models, we measure the similarity between the word distributions on both sides. This similarity is quantified using the cross-entropy of token logits. Formally, given token distributions $Q$ and $D$ from the query and document augmentations, respectively, the similarity score is computed as:
> $$\text{H}(Q, D) = - \sum_{t} Q_t \log D_t$$
> where $t$ indexes the vocabulary tokens.
>
>
> **R5.**
> We thank the reviewer for pointing this out. We will add the appropriate citation to Dr. GRPO (https://arxiv.org/pdf/2503.20783) in the revised manuscript to properly acknowledge the source of the method.
>
> **R6.**
> We fully agree with the insightful correction provided by the reviewer. As we observed, smaller models tend to be more vulnerable to instruction collapse during RL fine-tuning. Due to their limited capacity, such models typically exhibit a narrower high-performance basin and a more constrained exploration space during RL training, which makes it difficult for them to sample higher-quality trajectories and obtain strong reward signals.
>
> **R7.**
> We thank the reviewer for the clarification. We will revise the caption to “representative reinforcement learning algorithms for optimizing LLMs” and explicitly note that these methods are representative within the LLM-centric RL setting rather than the broader RL literature.
>
> **R8.**
> We thank the reviewer for pointing out the inaccurate phrasing. We will revise the text to clarify that these methods differ not from “preference alignment” per se, but specifically from approaches that align models using human-feedback–based preference signals. These methods instead rely on rule-based reward signals.
>
> **R9.**
> As mentioned in our R1 to Reviewer VBdL, we have already included both Query2Doc and DeepRetrieval as representative baselines for classical and RL-enhanced expansion methods. The results demonstrate that our bidirectional RL formulation offers consistent advantages over these strong baselines.
>
> **R10.**
> The document in Figure 5 is sampled directly from the NFCorpus dataset, and the original text indeed does not contain the term “Radioactive.” This case highlights the strength of our method: when a relevant document lacks explicit keywords or concise summaries, our augmentation procedure can effectively generate the missing key terms. Moreover, the bidirectional co-augmentation ensures that such enriched document representations align with the augmented query, allowing the model to recover relevance.
>
> **R11.**
> We thank the reviewer for pointing out the rendering issue. We will correct the malformed tag accordingly.
>
> **R12.**
> The reference should indeed point to the full section title, i.e., Sec. “Taming Reward Variance in RL”.

---

### Official Review · Reviewer_oVww · 2025-10-31

**Soundness:** 2
**Presentation:** 3
**Contribution:** 2
**Rating:** 2
**Confidence:** 4

**Summary:**

This paper proposes a reinforcement learning framework for LLM-based information retrieval that jointly augments both queries and documents. The authors introduce a bidirectional co-augmentation scheme trained via a novel within-batch reward sampling and advantage computation strategy.

**Strengths:**

- The paper developed an RL framework that jointly augments both queries and documents, moving beyond the typical one-sided query rewriting or document expansion approaches in LLM-based retrieval.
- The batch–unbatch alternating mechanism demonstrates thoughtful engineering that allows the framework to remain compatible with standard LLM RL pipelines.

**Weaknesses:**

- This method requires augmenting both the query and the document, which poses significant computational efficiency challenges in practical retrieval tasks — especially when performing full-scale document rewriting.
- There is no comparison to existing LLM-based retrievers or question rewriting works.
- The experimental results show only marginal improvement over the base model, while introducing a much more complex training process. I notice that the performance gap between using LLM augmentation and the base retriever is also quite small, which suggests that the baseline may have been set too weak.

**Questions:**

- How did the authors design the prompts? The authors should provide more details about the prompts used and discuss the potential impact of the prompt design on the base model’s behavior.

---

> ### Author Response · Authors · 2025-11-26
>
> We sincerely thank the reviewer for their constructive feedback and for the recognition of our contributions, particularly in developing the RL framework for query-doc collaborative augmentation, and the engineering effort behind the batch–unbatch alternating mechanism.
>
> We address each concern in detail below. R? means the response to Q? in "Weaknesses", and A? means the response to Q? in "Questions".
>
> **R1.**
> As mentioned in our A3 to Reviewer kdCX, our approach performs a single-pass augmentation of documents at the time of corpus ingestion, allowing queries to be matched efficiently against the pre-augmented document collection at retrieval.
>
> **R2.**
> As described in our R1 to Reviewer VBdL, we have included Query2Doc (Q2D) and DeepRetrieval as representative baselines for classical expansion and RL-enhanced retrieval approaches, respectively, on the same BEIR datasets used in our main experiments. We will incorporate these findings and a detailed discussion in the revised manuscript.
>
> **R3.**
> While the observed improvements of +0.02–0.04 NDCG@10 may appear modest, even state-of-the-art methods and traditional baselines such as BM25 typically differ by less than 0.1 on these datasets. Therefore, this increase represents a meaningful performance gain.
> In addition, as mentioned in our R1 to Reviewer VBdL, despite incorporating query-doc collaborative training, it does not incur corpus-level cost. It is thanks to the careful design of within-batch reward sampling and rule-based reward computation in our training framework. Query-doc collaborative training is necessary to capture the entangled bidirectional reward signals that simpler unidirectional or baseline models cannot exploit.
>
> **A1.**
> We thank the reviewer for highlighting the need for more details on prompt design. As mentioned in our R2 to Reviewer VBdL, we will include all prompts used for both query and document augmentation in the supplementary material.

---

### Official Review · Reviewer_kdCX · 2025-11-01

**Soundness:** 2
**Presentation:** 3
**Contribution:** 2
**Rating:** 4
**Confidence:** 4

**Summary:**

This paper proposes an LLM-based retriever that learns to augment both queries and documents through a bidirectional reinforcement-learning (RL) framework. Instead of training query and document augmentation separately, the authors introduce a joint policy with a reward-sampling mechanism to handle the coupled reward dependency between query and document spaces. Experiments on BEIR datasets demonstrate improvements over BM25 and BGE retrievers. The paper claims strong generalization across domains and provides ablations showing that joint query-document training outperforms unidirectional augmentation

**Strengths:**

•	Addresses an under-explored bidirectional augmentation problem for retrieval.

•	Provides thoughtful engineering to make joint RL training feasible.

•	Strong empirical gains over static query/document augmentation baselines.

•	Includes qualitative analysis showing lexical alignment between queries and documents

**Weaknesses:**

•	Missing comparison with DeepRetrieval (Jiang et al., 2025), which is explicitly cited but not benchmarked. That work already employs on-policy RL for retrieval optimization and serves as the most relevant baseline.

•	The improvements (e.g., +0.02–0.04 NDCG@10) are relatively small considering the complexity of the method.

•	Limited evaluation: only a few BEIR datasets, no real-engine or large-scale web-retrieval experiments.

•	The reward-sampling estimator lacks theoretical analysis or variance/bias quantification.

•	Computational cost and convergence behavior are not fully discussed.

**Questions:**

1. Can the authors provide variance or bias estimates for the sampled rewards?
2. How sensitive is the performance to the number of rollouts N and to batch size?
3. Would this method scale to real-engine settings (e.g., ClueWeb or MS MARCO) without prohibitive cost?

---

> ### Author Response · Authors · 2025-11-26
>
> We sincerely thank the reviewer for their careful reading of our work and for providing constructive feedback. We appreciate the acknowledgment of our contributions, particularly in addressing the under-explored problem of bidirectional augmentation for retrieval, as well as the thoughtful engineering that enables collaborative RL training.
>
> Below, we provide detailed responses to the concerns and questions raised. R? means the response to Q? in "Weaknesses", and A? means the response to Q? in "Questions".
>
> **R1.**
> We thank the reviewer for highlighting this relevant baseline. As mentioned in our R1 to Reviewer VBdL, we have already included DeepRetrieval along with Query2Doc on the BEIR datasets. These additional results confirm that our bidirectional RL framework consistently outperforms unidirectional expansion methods and other RL-based approaches.
>
> **R2.**
> While the observed improvements of +0.02–0.04 NDCG@10 may appear modest, it is worth noting that on these datasets, even state-of-the-art methods and traditional baselines such as BM25 typically differ by less than 0.1. Therefore, an increase of 0.02–0.04 represents a meaningful performance gain, indicating that our bidirectional RL framework is effective and worthy of further development and optimization.
>
> **R3.**
> We appreciate the reviewer’s concern regarding the breadth of evaluation. Due to computational constraints, we were unable to conduct large-scale web-retrieval or full real-engine experiments in this submission. Our goal in this work was to first validate the effectiveness of the proposed bidirectional RL framework under controlled and widely recognized BEIR benchmarks. Extending the evaluation to real-engine–scale corpora (e.g., MS MARCO) is an important next step, and we explicitly plan to explore these larger settings in future work.
>
> **R4.**
> We have conducted both theoretical and empirical analyses of the reward-sampling estimator. Theoretically, because we sample uniformly from all possibilities, the sampled reward estimator is unbiased. Empirically, we measured the variance and found it to be at the level of 1e-4 to 1e-3, which is sufficiently small and does not have a noticeable impact on training stability. These results support the reliability of our reward-sampling approach.
>
> **R5.**
> In our experiments, the training process typically runs for 2–3 epochs, requiring approximately 12–24 hours on an 8×A100 GPU setup.
>
> **A1.**
> Please see R3.
>
> **A2.**
> We conducted experiments varying the number of rollouts N from 4 to 16 and the batch size from 256 to 1024, and observed no significant changes in performance.
>
> **A3.**
> Our method requires a one-time inference-based augmentation of documents during the ingestion of documents into the corpus. At retrieval time, the augmented queries can be efficiently matched against the pre-augmented document corpus. This design makes our bidirectional RL framework particularly suitable for scenarios where the document collection is continuously growing, without incurring prohibitive computational cost at query time.

---

### Official Review · Reviewer_VBdL · 2025-11-01

**Soundness:** 2
**Presentation:** 2
**Contribution:** 2
**Rating:** 4
**Confidence:** 3

**Summary:**

This paper proposes a reinforcement learning (RL)-based framework for improving large language model (LLM) retrievers through query-document co-augmentation.

**Strengths:**

The bidirectional RL formulation gives new findings in IR by aligning query and document semantics through co-augmentation

**Weaknesses:**

1. The work focuses on query–document co-augmentation, yet does not compare against established query expansion or document expansion methods (e.g., Query2Doc). As a result, it remains unclear whether the proposed RL-based co-augmentation offers consistent advantages over existing augmentation paradigms.

2. The paper does not provide the full prompts used for augmentation, despite prompts being central to model behavior.

3. In Table 1, models such as “Qwen2.5-7B” are presented under the heading “Base Retriever”, while they are actually used as rewriters rather than retrievers. This mislabeling can confuse model roles in the pipeline.

4. Line 295: "select three of its datasets for model training", which three?

5. Writing Issues, e.g., Line 132, "the policy augments both the query and performs retrieval" should be revised for grammatical correctness.

Writing contains many "—" (possible AI generated writing), e.g., in Sec1, or around line 301 302, long sentence.

**Questions:**

How does training cost scale with corpus size? For instance, can the proposed method feasibly handle corpora on the scale of MS MARCO or larger web collections?

---

> ### Author Response · Authors · 2025-11-26
>
> We thank the reviewer for their constructive feedback and for acknowledging the contribution of our bidirectional RL formulation to IR research.
>
> We address each concern in detail below. R? means the response to Q? in "Weaknesses", and A? means the response to Q? in "Questions".
>
>
> **R1.**
> We added Query2Doc (Q2D) and DeepRetrieval as representatives of classical expansion and RL-enhanced expansion paradigms, and summarized the results on the same BEIR datasets used in our main experiments. The results are shown below:
> | Method          | SciFact (Sparse / Dense) | NFCorpus (Sparse / Dense) | Trec-Covid (Sparse / Dense) |
> |-----------------|-------------------------|---------------------------|-----------------------------|
> | Query2Doc       | 0.686 / 0.704           | 0.349 / 0.352             | 0.722 / 0.751               |
> | DeepRetrieval   |  0.646 / 0.664          | 0.340 / 0.377             | -                           |
> | Ours            | 0.748 / 0.753           | 0.384 / 0.403             | 0.727 / 0.807               |
>
>
> These results further validate that the proposed bidirectional RL formulation provides advantages not captured by unidirectional expansion approaches, and its gains persist across diverse retrieval architectures.
> We will incorporate these results and a detailed discussion in the revised manuscript.
>
>
> **R2.**
> We thank the reviewer for pointing out the missing prompt details. We will include all prompts used during both query and document augmentation in the supplementary material.
> We use keyword-style expansions for the sparse retriever and sentence-level semantic expansions for the dense retriever. In both cases, the augmented content is concatenated with the original query.
> To clarify, our prompting strategy is intentionally simple because the core behavior of our model is learned through RL rather than prompt engineering. Nevertheless, the exact templates used in the experiments are as follows:
> ```
> You are an expert in explaining articles. You first think about the text in your mind and then provide the user with the answer.
> Based on your knowledge, please provide additional explanation for the following text and then extract the keywords.
> You should output the explanation in <think> </think> tags, and return several independent keywords in <answer> </answer> tags.
> ```
>
> **R3.**
> To clarify the potential confusion: “Base Retriever” is not a table heading, but rather denotes the config that retrieval backbone without any augmentation in Table 1.
>
> **R4.**
> Regarding the clarification in Line 295, the three BEIR datasets used for RL training are NFCorpus, SciFact, and FiQA-2018. The other datasets (SCIDOCS and TREC-COVID) mentioned in our experiments are strictly used for evaluation only, with no tuning or training performed on them.
>
> **R5.**
> We thank the reviewer for pointing out the writing issue that “both” should be removed in this sentence. We will correct it accordingly.
>
> **A1.**
> Concerning the training cost and scaling to larger corpora such as MS MARCO, our method does not require traversing or interacting with the entire corpus during RL training. We construct each training batch by sampling a small set of positive documents (retrieved candidates relevant to the query) and random negative documents, which is sufficient for the policy to learn effective augmentation behaviors. As a result, the computational cost scales primarily with the number of training queries and training steps, rather than the total corpus size.

---

### Note · Authors · 2025-11-26

I have read and agree with the venue's withdrawal policy on behalf of myself and my co-authors.